# Efficient Discrete Element Modeling of Particle Dampers

Fabio Biondani [1], Marco Morandini [1,*], Gian Luca Ghiringhelli [1], Mauro Terraneo [2] and Potito Cordisco [2]

1   Politecnico di Milano, Dipartimento di Scienze e Tecnologie Aerospaziali, 20156 Milano, Italy; fabio.biondani@polimi.it (F.B.); gianluca.ghiringhelli@polimi.it (G.L.G.)
2   Vicoter, 23801 Calolziocorte, Italy; mauro.terraneo@vicoter.it (M.T.); potito.cordisco@vicoter.it (P.C.)
*   Correspondence: marco.morandini@polimi.it

**Abstract:** Particle dampers' dissipative characteristics can be difficult to predict because of their highly non-linear behavior. The application of such devices in deformable vibrating systems can require extensive experimental and numerical analyses; therefore, improving the efficiency when simulating particle dampers would help in this regard. Two techniques often proposed to speed up the simulation, namely the adoption of a simplified frictional moment and the reduction of the contact stiffness, are considered; their effect on the simulation run-time, on the ability of the particle bed to sustain shear deformation, and on the prediction of the dissipation performance is investigated for different numerical case studies. The reduction in contact stiffness is studied in relation to the maximum overlap between particles, as well as the contacts' duration. These numerical simulations are carried out over a wide range of motion regimes, frequencies, and amplitude levels. Experimental results are considered as well. All the simulations are performed using a GPU-based discrete element simulation tool coupled with the multi-body code MBDyn; the results and execution time are compared with those of other solvers.

**Keywords:** particle damping; discrete element method; GPU computing; energy dissipation; contact stiffness

## 1. Introduction

Particle dampers are passive damping devices consisting of one or more enclosures partially filled with particles. They are typically mounted on vibrating deformable systems to increase damping and limit structural vibrations. These devices can achieve high energy dissipation rates by means of inelastic collisions and friction between particles and between the particles and the enclosure's inner wall.

Thanks to their simplicity and versatility, particle dampers have been considered for a large number of use cases in the literature, ranging from gas turbine and compressor blades to attenuation in aeronautical honeycomb panels and even earthquake isolation in buildings [1,2].

The interest in these devices comes from their unique characteristics: they are fairly simple, are passive devices, are suitable for application in harsh environments, and are not very sensitive to high temperatures; furthermore, they can be easily retrofitted in existing structures and can be embedded in systems in a non-obstructive way by taking advantage of preexisting cavities. Particle dampers can be manufactured in multiple ways, an extreme example being additive manufacturing, with a high number of particles sized a few micrometers [3–5].

An interesting and comprehensive introduction to particle dampers is reported in [2].

The strong influence of the inner motion regime of the particles on the amount of achievable dissipation has been demonstrated numerically and experimentally [6,7]. The particle motion regimes can range from a solid bouncing agglomerated behavior of the particles to local or global motions resembling that of a fluid. The dissipation characteristics of the simplest motion regime, denominated "bouncing bed", can be modeled with simple theories such as Friend and Kinra's theory [8]; however, even in this working condition,

particle dampers show, for many applications, hard-to-predict highly non-linear dynamic characteristics. Furthermore, as can be expected from the notorious bouncing ball problem [9,10], particle dampers' response to oscillations is not guaranteed to be periodic and can be chaotic [11].

Literature studies show that particle dampers have been analyzed and designed to find the best performance by varying a limited number of parameters (such as filling ratio and particle size [12]), either resorting to experimental-only approaches or numerical and analytical methods. Regarding the latter, analytical methods include models that consider the particle bed as an agglomerated bouncing solid body, linearization of the particle damper damping around certain conditions, or other simplified models based on the multiphase flow theory.

Analytical methods are often limited to specific cases and may be difficult to calibrate without experimental data [2]; therefore, particle dampers analyses are often based on numerical methods, which may be easier to calibrate. Implementations of the discrete element method (DEM) are often chosen, sometimes in co-simulation with finite element or multibody solvers.

Despite the broad versatility of the discrete element method, the major concern in its application to particle dampers is its high computational cost, mainly due to the need to use very small simulation time steps. More efficient discrete element method simulations would assist in the design of particle dampers, as their design could require the analysis of a large number of conditions; e.g., the application of particle dampers on aeronautical components would require the dampers to work effectively on conditions characterized by different excitation intensity, frequency content, and direction and intensity of inertial forces during maneuvers, while containing the weight increase. Additionally, other fields may benefit from an increased simulation efficiency, e.g., the simulation of particle dampers realized by means of additive manufacturing [13].

Discrete element method codes significantly increased their efficiency from the initial penalty-based formulation developed by Cundall and Strack [14]. Efficient broad-phase and narrow-phase algorithms for contact detection significantly reduced the contact detection phase run time [15,16]. Furthermore, alternative non-smooth formulations were developed; such formulation, such as that implemented by Tasora in the solver Chrono [17], uses complementarity conditions to enforce non-penetration of the discrete elements that come into contact [18], and can allow time steps that are significantly larger than those required by traditional penalty-based methods. It is worth noting that the non-smooth formulation implemented in Chrono has also been employed to analyze particle dampers using a tight coupling scheme for multibody co-simulation [19,20].

Regarding penalty-based methods, significant reductions in the overall computational time have recently been obtained by taking advantage of modern graphic computing units (GPU) computing techniques. Blaze-DEMGPU [21] and Chrono::GPU [22] are two examples of solvers employing this technology; the latter is also characterized by innovative solutions to use complex contact models that require memory of the past contacts, with complex GPU programming to increase the floating-point precision and operation speed via a transformation of the coordinates.

An approach often proposed to speed up the analyses based on penalty-based DEM formulations is to lower the contacts' normal and tangential stiffness. Coetzee [23] mentioned that reducing the contact stiffness by a factor of up to 1000 is a common method that allows computation time to be reduced, but care has to be taken not to exceed a reasonable particle overlap. A relation between contact stiffness and interference for a Hertz contact model was suggested by Malone [24]; he suggested that, given a maximum allowable overlap and an estimation of the maximum speed of the particles, the proper value of stiffness can be defined. Lommen found that, for bulk compression, angle of repose, and penetration tests, normal overlaps should be kept below 0.3% of the particle radius. Clearly [25], who analyzed granular flows, stated that typically average overlaps of 0.1–1.0% are desirable. Methods to determine the normal stiffness as a function of the

maximum dimensionless overlap and the dimensionless contact duration were developed by Navarro [26]; it is worth noting, however, that such stiffness depends on the particles' impact velocity that, in principle, is not known a priori.

No reference to the adoption of the aforementioned approaches was found in the literature regarding particle dampers. Only two proposals to speed up DEM simulations of particle dampers were found. These two studies relied on simple configurations characterized by strict alignment constraints and a "bouncing bed" regime of motion of the particles. In the first one Sanchez [27] showed that the particle dampers' performance does not depend on the material's contact dissipative properties, thanks to the "inelastic collapse" of the particles, and that, in optimal bouncing conditions, the analysis of the particle damper using a single agglomerated particle with null restitution coefficient would provide the same results. In the second one, Saeki [28] proposed the partition of a vertical moving damper and the substitution of the particles with larger ones (a technique known as agglomeration).

In this paper, DEM techniques to manipulate the contact stiffness and the friction models are developed, implemented, and used to simulate particle dampers. The scope of this work is to quantify the impact that such techniques have on execution time and accuracy of the analyses, with the ultimate goal of developing an efficient and reliable tool to be used in the design of particle dampers.

Most of the simulations are carried out using a recently developed GPU penalty-based discrete element solver that is tailored to the simulation of particle dampers; the code's contact models and the adopted co-simulation technique with the multibody code MBDyn [29] are described. The performance of this code is compared with other DEM solvers, for simulations with a number of spherical particles ranging from a hundred to more than a million, performed on a consumer-grade hardware; the longest of these simulations lasted more than two days. Analyses of different particle damping case studies are performed both with a simplified and a traditional tangential contact model, with hundreds of particles, and size ranging in the millimeters. The ensuing results are compared for a wide range of excitation levels, for frequencies from 5 Hz up to 100 Hz. The simplified tangential contact model is a simple directional friction model neglecting static friction; this approach provides significant computational advantages, and therefore it is worth analyzing in which condition it can be used in the analysis of particle dampers.

A novel technique to modify the particles' contact stiffness is proposed as well, and its performance is evaluated in terms of accuracy and cost efficiency; the validity of the considered techniques and their impact on the results are discussed for different excitation levels and motion regimes, analyzing a case study from the literature.

A new experiment, a cantilever beam with a particle damper positioned near the free end, is considered. Its dynamics are reproduced in co-simulation with MBDyn.

## 2. Materials and Methods

This section describes the DEM simulation code PMB (Particles with Moving Box) and its coupling with the multibody code MBDyn. The software mentioned above was selected because MBDyn is a general-purpose open-source multibody solver [29] which can be easily integrated into co-simulation schemes, while PMB is a proven and efficient DEM code [30]. Both codes are developed by the authors, and thus allow seamless integration and modification.

A technique to scale the normal contact stiffness in particle damping simulations, with the aim of saving computational time without significantly affecting the results, is presented.

To compare the damping performance of particle dampers, a procedure to estimate the particle damper's equivalent damping is proposed. Finally, the description, dimensions, and procedure of the in-house experiment are reported.

*2.1. PMB: Discrete Element Method Solver Tailored to Particle Dampers*

The PMB software is a GPU-based tridimensional solver for particle dampers with spherical particles of uniform radius. The basic structure of the software was inherited from the sample program "particles" [31] provided within CUDA's sample programs, which was modified to allow for arbitrary movement of the external box; hence, the software was called PMB (Particles with Moving Box).

To analyze the effect of particle dampers on deformable systems, PMB was coupled with MBDyn.

### 2.1.1. Algorithms Description

Broad-phase particle-to-particle contact detection is performed with a loose uniform grid spatial subdivision and sorting [15,32] and a neighboring-cell contact detection scheme [33]; this is basically the same method used by the "particles" CUDA example code, slightly modified to consider the motion of the particle damper's enclosure.

Since PMB considers only monodisperse spherical particles, the above simple scheme suffices; there is no need to implement more complex and costly broad-phase algorithms. In order to employ bit-wise operations to quickly assess in which cell a particle is, the grid size is always a power of two. Analyses of certain case studies showed that choosing a cell grid side that is about twice the particles' radius minimizes the execution time; this value agrees with Mio's results [34].

After the broad-phase, each particle is checked for collisions with the enclosure's walls and with the particles in the neighboring grid cells. If collisions are detected, the contact forces and moments are computed based on the relative overlap and the relative velocity between the colliding particles in the contact point.

The normal force between two particles (or between a particle and a wall) is computed with a linear spring-dashpot (LSD) model. The literature shows that the value of that spring is usually determined by experience or guessed so that desirable values of overlap between particles are obtained [25]; alternatively, it can be computed from the material properties, such as Young's modulus, Poisson's ratio [33], and yield strength [35]. The damping coefficient is determined from the restitution coefficient using a model from the literature [25,36,37].

After the contact forces are computed, PMB solves both the rotational and linear equations of motion for each particle using the so-called semi-explicit Euler time integration scheme (also known as symplectic Euler [38]), an explicit integration scheme for second-order equations. Since the particles are spherical their orientation does not directly influence the equations of motion; therefore the particles' orientation is not computed, and the code tracks only the particles' angular velocity. If required, the particles' orientation can be optionally integrated by resorting to quaternions.

Since explicit integration schemes are only conditionally stable, the time step is determined using Hart's formula [39]:

$$\Delta t = \alpha\, 2 \sqrt{\frac{m_p}{2 k_p}}, \tag{1}$$

where $m_p$ and $k_p$ are the mass and the maximum normal contact stiffness of a particle, and $\alpha$ is a constant that should be as low as 0.1 to ensure numerical stability, although a value of 0.2 is generally sufficient.

For each time step, PMB can efficiently compute the following quantities, related to the overall behavior of the particles, through parallel reduction:

- The total contact force and moment acting on the particle damper's enclosure walls;
- The particles' overall center of mass and inertia tensor;
- The maximum contact overlap;
- The granular temperature, as defined by Campbell [40], and mean particles velocity;
- The dissipated energy in the impacts computed by integrating the dissipated power.

Different analytical surfaces can be combined in order to define the enclosure shape; each surface can be independently translated and rotated in 3D.

### 2.1.2. Friction Models

Since the particles are in motion when the particle damper is working in damping conditions, as a tangential contact force model, the authors initially used a highly simplified Coulomb's law of friction that neglects static friction, shown in the following equation:

$$
\begin{cases}
\mathbf{f}_T = -\mu_f |\mathbf{f}_N| \frac{\mathbf{v}_T}{|\mathbf{v}_T|} & \text{if } |\mathbf{v}_T| > 0 \\
\mathbf{f}_T = \mathbf{0} & \text{otherwise } (|\mathbf{v}_T| = 0),
\end{cases}
\tag{2}
$$

where $\mathbf{f}_T$ and $\mathbf{f}_N$ are the exchanged tangential and normal forces, $\mu_f$ is the friction coefficient and $\mathbf{v}_T$ is the relative tangential velocity between the contacting bodies in the contact point. This model is deemed as "insufficient" by Matuttis and Chen [41], since it is unphysical and unable to reproduce the simplest problems for static friction. They suggested that using elastic-frictional models, such as one based on the Mindlin–Deresiewicz model [42], would be more appropriate for DEM simulations. Nevertheless, this simple model was also used in other known particle-damping-related works, e.g., [43] and Fleissner [44], who suggested that, in highly dynamic systems, slipping friction is dominant, and it is often legitimate to neglect sticking friction. Furthermore, the described friction model has a significant computational advantage over more complex models with static friction, as it does not need to store and use internal state variables, related to the tangential overlap, for the contacts a body is having.

Preliminary simulations, compared with in-house experiments, allowed us to verify that the simple friction model is not able to correctly reproduce the local fluidization motion regime of the particles, while it leads to overall good results when the particles keep moving during a period; this makes sense, since the local fluidization involved a transition from static to dynamic friction, and static friction is not accounted for with the simple friction model. Because of this, the ability to store the contacts between bodies and the contact models' internal variables, within a time step. This allows for the implementation of a tangential contact model accounting for static friction. The tangential friction force $\mathbf{f}_T$ is defined by a tangential LSD model whenever the module of the tangential force is lower than the friction limit (the contact is in a pre-sliding condition) and by a constant friction force otherwise (the contact is sliding),

$$
\begin{cases}
\mathbf{f}_T = -k_T \mathbf{u}_T - c_T \mathbf{v}_T & \text{if } |k_T \mathbf{u}_T + c_T \mathbf{v}_T| \leq \mu_f |\mathbf{f}_N| \\
\mathbf{f}_T = -\mu_f |\mathbf{f}_N| \frac{\mathbf{v}_T}{|\mathbf{v}_T|} & \text{else if } |\mathbf{v}_T| > 0 \\
\mathbf{f}_T = -\mu_f |\mathbf{f}_N| \frac{\mathbf{u}_T}{|\mathbf{u}_T|} & \text{otherwise } (|k_T \mathbf{u}_T + c_T \mathbf{v}_T| > \mu_f |\mathbf{f}_N| \text{ and } |\mathbf{v}_T| = 0),
\end{cases}
\tag{3}
$$

where $k_T$ and $c_T$ are the constants defining the tangential LSD, and $\mathbf{u}_T$, the tangential relative displacement at the contact point, is the internal variable of the contact model. The tangential relative displacement is initialized to a null vector, $\mathbf{u}_T = \mathbf{0}$, at the onset of the contact, and updated at every time step according to

$$
\mathbf{u}_T = \mathbf{u}_{T,\text{prev}} + \mathbf{v}_T \Delta t,
\tag{4}
$$

where $\mathbf{u}_{T,\text{prev}}$ is the tangential relative displacement at the contact point at the previous time step; should the contact be in relative sliding, i.e., $|k_T \mathbf{u}_T + c_T \mathbf{v}_T| > \mu_f |\mathbf{f}_N|$, the tangential relative displacement is later re-computed by inverting the first of Equation (3) with the friction force $\mathbf{f}_T$ computed from the second or the third part of Equation (3):

$$
\begin{cases}
\mathbf{u}_T = ((\mu_f |\mathbf{f}_N| - c_T |\mathbf{v}_T|)/k_T) \frac{\mathbf{v}_T}{|\mathbf{v}_T|} & \text{if } |\mathbf{v}_T| > 0, \\
\mathbf{u}_T = \mu_f |\mathbf{f}_N|/k_T \frac{\mathbf{u}_T}{|\mathbf{u}_T|} & \text{otherwise.}
\end{cases}
\tag{5}
$$

Rolling friction has been included as well: a torque model with an elastic–plastic LSD was included following Holmes' work [45]. The rolling resistance torque is defined as:

$$\begin{cases} \mathbf{m}_R = -\mu_R R_r |\mathbf{f}_N| \frac{\boldsymbol{\omega}_{REL}}{|\boldsymbol{\omega}_{REL}|} & \text{if } |\boldsymbol{\omega}_{REL}| > 0 \\ \mathbf{m}_R = \mathbf{0} & \text{otherwise,} \end{cases} \tag{6}$$

where $\mu_R$ is a constant, $R_r$ is the effective radius, and $\boldsymbol{\omega}_{REL}$ is the relative angular velocity of the bodies in contact.

It is worth noting, however, that while the importance of rolling friction is clear in the simulation of sandpile formation [46], its importance has not been investigated in particle dampers, to the authors' knowledge; many studies that did not consider rolling friction were also validated with experimental results [6,43].

### 2.1.3. Co-Simulation with MBDyn

Coupling between PMB and MBDyn was achieved by setting an interprocess communication scheme based on sockets. The particle damper is represented as a node on MBDyn's side: MBDyn sends the position, velocity, rotation, and angular velocity of the particle damper's node to PMB, and receives the total force and moment resultants acting on the node, computed from the collisions of the particles with the enclosure walls. The coupling scheme, shown in Figure 1, is loose (non-iterative), and it was designed to parallelize the calculations performed by the two coupled software as much as possible; this allows us to significantly decrease the execution time whenever the simulation time required for the multi-body model comparable to that of the particle code (i.e., for modes with few particles or, alternatively, significantly complex multibody models).

A loose co-simulation scheme does not require iterations of the communications within a time step; hence, it is computationally less expensive than a tight scheme. A tight scheme would have provided significant numerical stability advantages in case of large time steps, as shown by Zhang when coupling MBDyn to Chrono non-smooth [19]; nevertheless, a loose scheme proves to be sufficient to couple MBDyn to a smooth DEM solver, such as PMB, because a small time step is imposed by the DEM solver explicit integration algorithm.

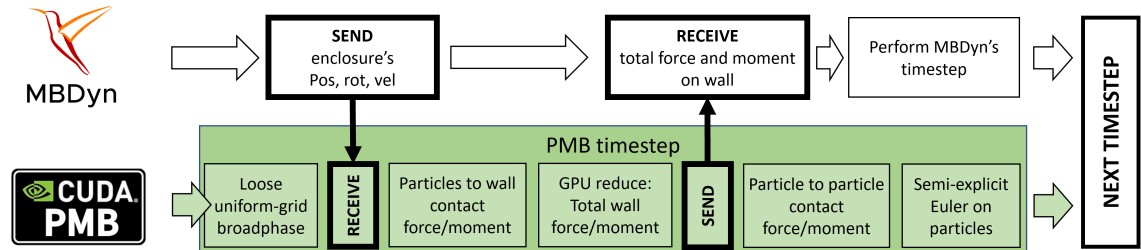

**Figure 1.** MBDyn–PMB coupling scheme.

### 2.2. Contact Stiffness Sizing Technique

The normal contact stiffness has a major impact on the execution times of DEM simulations. Since the biggest drawback in employing a DEM is its high computational cost, analyzing the impact of the variation of the contact stiffness on the accuracy of the analyses can lead to a significant speed increase. The normal contact stiffness can be determined from the colliding bodies' material properties [33]. Reducing the contact stiffness from these values is a common method to speed up the analyses without significantly affecting the results, as long as a reasonable particle overlap is not exceeded [23]; that is to say that, as long as the stiffness is not reduced below a certain value, the sensitivity of the results of DEM analyses to a variation in contact stiffness is negligible.

Many papers [23,25,47,48] consider the relation between either the maximum or the average overlap in penalty-based DEM codes and the accuracy of the analyses, and report,

for specific applications [23,25,47,48], the adopted values of the permissible overlap ratios or of the contact stiffness reduction. Methods to scale the contact stiffness of an LSD model given a certain maximum overlap, a certain contact duration, or a certain mean overlap were also developed by Navarro [26].

It has to be remarked that decreasing the contact stiffness is just an artifice to reduce the execution times and that a contact using softened parameters leads to a model that approximates what would happen with the stiffness parameters adjusted to mimic the actual material stiffness of the particle. However, when optimizing the design of a particle damper with DEM simulations for a given mechanical system, there may be the need to perform a large number of simulations varying several particle dampers parameters, e.g., enclosure shape, size, orientation, or particle size, regardless of the objective of the design optimization. For this reason, reducing the overall computational time as much as possible can be important, provided that the overall response trend for varying parameters is faithfully reproduced. The aim is twofold: for a given hardware setup, either increase the overall number of particles that can be simulated once within a reasonable time (say, two days for each simulation and working condition) or increase the number of particles for which one can reasonably perform an optimization of the particle damper performance (with, say, an hour required for each simulation). To this end, the attention is focused onto the effect of the contact stiffness on global performance indices, such as the dissipated energy per cycle, the particles' overall center of mass position or moments of inertia and, for a particle damper mounted on onto a deformable structure, the amplitudes of the response's motion of the structure. Local responses, such as the position of a given particle, are not compared for different contact stiffness, as they are neither a quantity of interest in the design of the damper, nor relevant in the global behavior of the particles' bed.

In this section, a formula to estimate the minimum value of normal contact stiffness that limits the maximum overlap of the impacts and does not affect the simulation results is proposed:

$$k_{n,\min} = m_p \left( \frac{\beta\, v_{PD,\max}}{\eta\, r_P} \right)^2, \tag{7}$$

where $k_{n,\min}$ is the minimum allowable normal contact stiffness, $m_p$ and $r_p$ are the mass and radius of one particle, and $v_{PD,\max}$ is the maximum velocity of the particle damper enclosure. As explained below, the parameters $\beta$ and $\eta$ are determined from the results of a series of short simulations performed on a set of particle dampers configurations of interest, repeated for different values of normal contact stiffness ranging several orders of magnitude.

For a given simulation $i$, the parameter $\eta_i$ is defined as the ratio between the maximum overlap $s_{\max,i}$ and the particle's radius $r_{p,i}$,

$$\eta_i = s_{\max,i} / r_{p,i}.$$

The parameter $\beta_i$ is defined as the ratio between the maximum particle overlap $s_{\max,i}$ and the overlap a pure elastic particle with contact stiffness $k_{n,i}$ would experience when impacting a rigid wall at velocity $v_{PD,i}$, the maximum velocity in simulation $i$,

$$\beta_i = s_{\max,i} / s_{REF,i}$$

where

$$s_{REF,i} = \sqrt{\frac{m_{p,i}}{k_{n,i}}}\, v_{PD,i}. \tag{8}$$

The global quantities of interest, obtained for each short simulation, are compared with the corresponding quantities obtained by using the nominal contact stiffness. The simulations for which the error is greater than a given, user-defined threshold are discarded, while all the remaining simulations are retained as acceptable. As a first step, the maximum overlap $s_{\max}$, computed over all the acceptable simulations, allows us to define the threshold

value of $\eta$ as $\eta = \max_i(s_{\text{max},i}/r_{p,i})$. This value of $\eta$, in turn, trivially corresponds to a given maximum overlap $s_{\text{max}} = \eta r_p$, where all quantities are now relative to the simulation that maximizes $\eta_i = (s_{\text{max},i}/r_{p,i})$; this overlap is set equal to the reference overlap that would occur by impacting a rigid wall, $\eta r_p = \beta s_{\text{REF}}$. By using $s_{\text{REF}} = \eta r_p/\beta$ within Equation (8), and solving it for $k_n$, one gets

$$k_n = m_p \left( \frac{\beta v_{PD}}{\eta r_P} \right)^2. \tag{9}$$

The minimum normal stiffness $k_{n,\text{min}}$ is now chosen conservatively from Equation (9) by using the maximum value of $\beta$ computed over all the acceptable simulations, $\beta = \max(\beta_i)$, instead of the actual value of $\beta$ from the simulation that maximized $\eta$.

Note that decreasing the contacts stiffness affects the duration of the contacts; hence, care must be taken when this method is blindly applied, especially to particle dampers subjected to high-frequency motions; in such instances, the normal contact stiffness should be a function of the contacts durations, and not of the maximum overlap.

Figure 2 shows the speed-up ratio obtained by using the most convenient criterion to size the normal contact stiffness in a sine-oscillating particle damper with the constraints of a max overlap given by $\beta = 3$ and $\eta = 5\%$, and a contact duration at least 100 times lower than the oscillation period. The speed-up ratio was computed with reference to 3 mm diameter steel particles. The contour lines are arranged in concentric inverted v's; the left branch of the v's is determined by the maximum-overlap criterion, while the right branch is determined by the contact duration criterion. Therefore, the figure shows that a maximum-overlap criterion is best suited for lower frequencies of oscillation of the particle damper, while a contact duration criterion is best suited for higher frequencies. As anticipated, there are areas in the figure's domain where it would not be convenient to reduce the contacts' stiffness. For accelerations of few gs, and frequencies between 5 Hz and 10 Hz, which are those typical of some of the most cited works on particle dampers (e.g., [8,43]), speed-up ratios up to 16 can be achieved.

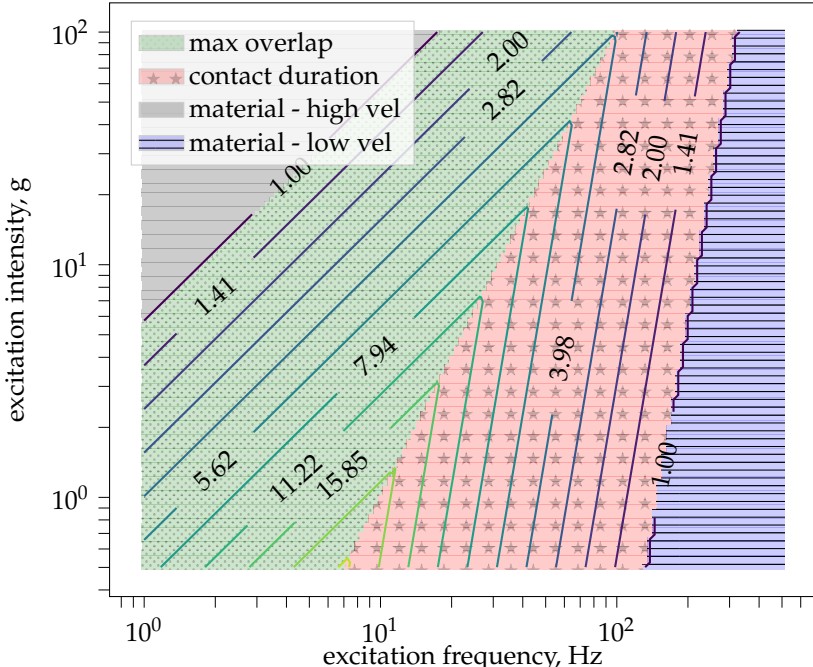

**Figure 2.** Contour lines plot of the speed-up ratio obtained by using the most convenient criterion to size the normal contact stiffness in a sine-oscillating particle damper with the constraint of $\beta = 3$, $\eta = 5\%$, and a contact duration at least 100 times lower than the oscillation period. The increment from a contour line and the next is exponential. Different fills have been used for the criteria mentioned in the legend.

### 2.3. Particle Damper Equivalent Damping Ratio

An index of the dissipation performance of a particle damper, the equivalent damping ratio $\xi_{PD}$, is proposed in this section. The equivalent damping ratio $\xi_{PD}$ can be useful, e.g., when designing a particle damper by resorting to using numerical simulations that allow computing the amount of dissipated energy, because it allows us to quantify and compare the effect of different particle dampers on the overall dynamics of a system.

When a particle damper is subjected to a sinusoidal motion of frequency $f_i$, we define the equivalent damping ratio $\xi_{PD}$ as the damping ratio of a mass-spring-dashpot one degree of freedom (DOF) system with natural frequency $f_i$ under the following assumptions:

- The system dissipates the same amount of energy per period of the particle damper;
- The system oscillates with the same sinusoidal motion of the particle damper;
- The total mass of the particles in the particle damper is $\mu$ times the lumped mass of the 1 DOF system. Here, $\mu$ is the mass ratio between the total mass of the particles and the mass of the system employing the particle damper.

Therefore, the dissipated energy of the 1 DOF system in an oscillation cycle can be expressed as:

$$E_{d,1T} = \int_{T_i} c_{1dof}\,\dot{x}^2\,dt = c_{1dof}\,\frac{A_V^2}{2\,f_i} = 2\,\xi_{PD}\sqrt{k_{1dof}\,m_{1dof}}\,\frac{A_V^2}{2\cdot\frac{1}{2\pi}\sqrt{k_{1dof}/m_{1dof}}} = \xi_{PD}\,\frac{m_p}{\mu}\,2\pi\,A_V^2, \tag{10}$$

where $E_{d,1T}$ is the energy dissipated per period; $k_{1dof}$, $m_{1dof}$, and $c_{1dof}$ are the stiffness of the spring, the mass, and the dashpot constant of the 1 DOF system, respectively; $A_V$ is the maximum velocity of the sinusoidal motion; $m_p$ the total mass of the particles; and $\xi_{PD}$ the equivalent damping ratio of the particle damper. The latter can be expressed as:

$$\xi_{PD} = \mu\,\frac{E_{d,1T}}{2\pi\,m_p\,A_V^2}. \tag{11}$$

A reference value to assign to $\mu$ can be equal to 1%; in such case, $\xi_{PD}(1\%)$ would represent the damping ratio that a system with a particle damper would have with a mass ratio $\mu$ equal to 0.01.

The assumptions taken in the definition of $\xi_{PD}$ do not always hold. According to the definition in Equation (11), the damping ratio of a particle damper system is linear with respect to its mass ratio; however, this is really not the case. In fact, as the mass ratio of the particle damper increases, the assumption of sinusoidal motion is increasingly false. In reality, the relation between $\xi_{PD}$ and $\mu$ should be linear for $\mu$ approaching zero and sub-linear otherwise, as shown by Cempel and Lotz in their theoretical studies over their definition of the loss factor of a particle damper [49]. One should also remember that adding a particle damper changes the overall mass of the system, hence its natural frequencies.

### 2.4. Experimental Validation

This subsection describes the experiments performed, their numerical simulation, and the validation of the code with the experimental results.

The experiment consisted of exciting a deformable system, coupled with a particle damper, with stepped sines at frequencies around the system's first modal frequency and measuring the response RMS. The amplitudes of the input sines were controlled to impose the level of the acceleration's RMS amplitude near the aforementioned particle damper.

Particle dampers are highly non-linear devices; for this reason, the transfer function of the system, measured between the input force and the acceleration near the particle damper, changes depending on the level of acceleration that the damper is subjected to. The measurements were therefore repeated for different levels of the imposed acceleration, ranging from 0.5 g to 2.5 g, with steps of 0.25 g.

The final result of the experiment is a relation between imposed acceleration and modal damping; such relation should resemble the relation between the dimensionless acceleration amplitude of the particle damper and the specific damping capacity shown by Friend and Kinra [8].

### 2.4.1. Setup of the Experiment

The experimental setup consists of a rectangular-section aluminum alloy cantilever beam with a cylindrical particle damper positioned near the free end. The cylinder's axis is parallel to the shorter side of the section. The cantilever beam's axis is horizontal, and the longer side of the rectangular section is vertical, parallel to gravity. A shaker excites the beam to vibrate horizontally, in the direction parallel to the beam cross-section's smaller side.

Four PCB 333B32 accelerometers were placed along the cantilever beam's length; the shaker's input force was measured using a PCB 208C02 load cell. The experimental RMS amplitude control on the particle damper was achieved by employing the acceleration measured near the particle damper as the test signal; the experimental control variable is the amplitude of the signal passed to the shaker. The acquisition was performed using an LMS-Scadas 316 system driven by the program TestLab. Slow-motion videos, capturing the experiment both from above and from the side, were recorded to analyze the particles' motion regimes at different levels of excitation.

The base and the top of the particle damper's enclosure are made with aluminum alloy, while the cylinder is made with transparent PMMA; steel particles were used.

The cantilever beam was subjected to stepped sines from 2 Hz to 10 Hz (around the first bending modal frequency), with 0.02 Hz steps, for every level of imposed acceleration considered. The transfer functions between the input force and the accelerometers' response were thus computed for every excitation level.

Figure 3 shows the experiment's beam, the shaker's and accelerometers' positions, and the placement of the cylindrical particle damper, with the cylinder axis parallel to the shaking direction.

Table 1 reports the system dimensions, positions, and material characteristics of the experiment's setup. The dimensions of the particles and the enclosure were determined by availability and time constraints; low oscillating frequencies were analyzed because literature references are more abundant for this regime, therefore more data to confirm the experimental trends could be used. The dimensions of the beam were also determined to obtain a significant mass ratio (about 5%) between particles' mass and structural mass and, consequently, a significant damping. The mechanical properties of the beam were obtained by updating the MBDyn beam model so that it matched the experimental response obtained with a beam without particles but with an equivalent lumped mass.

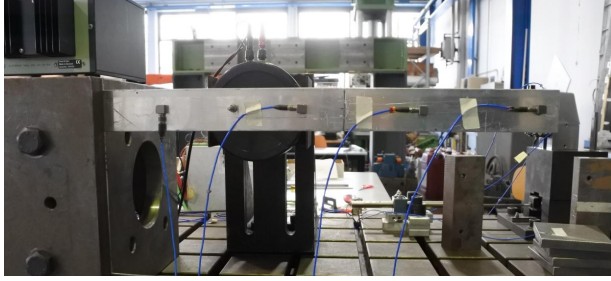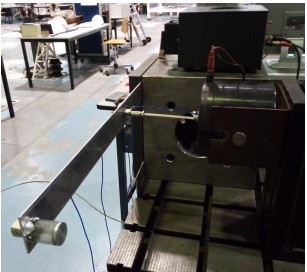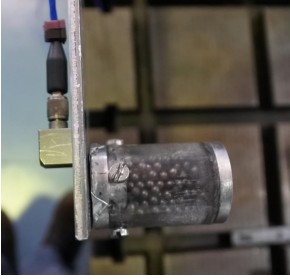

**Figure 3.** Experimental setup of the beam, the sensors and the particle damper; the particle damper axis is parallel to the beam's smaller side of the rectangular section.

**Table 1.** Experiment measured dimensions.

| Description | Value | | Description | Value | |
|---|---|---|---|---|---|
| cantilever beam length | 0.543 | m | shaker distance from root | 0.16 | m |
| beam section longer side | 0.05 | m | accelerometer #1 distance from root | 0.075 | m |
| beam section shorter side | 0.003 | m | accelerometer #2 distance from root | 0.241 | m |
| beam density | 2695 | kg/m$^3$ | accelerometer #3 distance from root | 0.382 | m |
| beam Young's modulus | 72 | GPa | accelerometer #4 distance from root | 0.519 | m |
| beam shear modulus | 27 | GPa | damper axis distance from root | 0.533 | m |
| beam structural damping | 0.5 | % | number of particles | 120 | |
| particles' material density | 7800 | kg/m$^3$ | particles diameter | 0.003 | m |
| empty particle damper mass | 0.0238 | kg | enclosure inner height | 0.025 | m |
| load cell mass | 0.030 | kg | enclosure inner diameter | 0.019 | m |
| mass of one accelerometer | 0.005 | kg | enclosure bottom plate thickness | 0.01 | m |

2.4.2. Modeling the Experiment Using MDByn

The cantilever beam was modeled in MBDyn using "beam3" elements, i.e., geometrically exact non-linear finite volume beam elements with three nodes, characterized by a formulation that is intrinsically free from shear locking [50].

The transfer function for a given imposed acceleration amplitude was obtained by reproducing the amplitude-control technique for the coupled multi-body and particle damper system for frequencies ranging from 5 Hz to 10 Hz. In order to control the acceleration RMS, an automatic iterative procedure allows us to change the sinusoidal external force magnitude, at each and every frequency, until the sought RMS of the response is achieved within a given tolerance for a given number of consecutive periods. This procedure is computationally expensive; in fact, for each input frequency and target imposed amplitude, it iterates by changing the forcing term amplitude, but needs to wait until the response RMS is stabilized before moving to the next iteration.

Since the convergence of the response is often made difficult by strong non-linearities, some strategies were adopted to speed up the simulation. The most effective ones were

- To use a large tolerance on the convergence values; this may be required because a particle damper is a device that consists of distinct impacting bodies that may impact with slightly different dynamics from one cycle to the next, thus changing the response;
- To save time by adjusting the input before the response amplitude converged if the response is too low or too high without any reasonable doubt;
- To change the strategy used to compute the next guess of the input amplitude during the analysis; two strategies are used to compute the next guess input amplitude:
  - Employ a "loosen" gradient descent method, in which the gradient is computed accounting for the large tolerance on the convergence;
  - Multiply the previous input amplitude by the ratio between the values of the target RMS response and the converged RMS response;
- To recognize two-periods limit cycles;
- In case of failure of the previous strategies, the stabilized result that gave the value closer to the target value is assumed. In these circumstances, the system is likely to not have a converging response.

The last two strategies were implemented because particle dampers are highly non-linear devices, and their response is not guaranteed to be periodic; rather, it can be chaotic or can exhibit limit cycles [11].

**3. Results**

*3.1. Timings and Results Comparison between Solvers on Simple Case Study*

In this section, PMB's performance is compared with other available DEM solvers. A simple case study was selected and analyzed with PMB, Abaqus Explicit, Chrono, and the GPU version of Chrono. Regarding Chrono, the "PARALLEL" module was used to

exploit Chrono's multicore capability, and both the available smooth and non-smooth DEM methods, called SMC and NSC, were employed. Abaqus Explicit's DEM solver had to be run in serial. PMB's performance was evaluated for both the implemented friction models (directional and elastic-frictional with static friction).

The case study is a cylindrical particle damper whose motion along its axis is a sine of imposed amplitude and frequency. Gravity is aligned to the particle damper's motion. The initial condition of the spherical particles is an FCC lattice leaning on the flat bottom floor of the cylindrical cavity. The initial velocity of the particles equals that of the enclosure, whose motion is sinusoidal.

The analyses were repeated with varying numbers of particles and particle radius, with a constant filling ratio of the cylinder. Table 2 reports the case study parameters.

**Table 2.** PMB validation: case study dimensions.

| Description | Value | |
|---|---|---|
| motion frequency | 10 | Hz |
| motion amplitude | 0.01 | m |
| gravity | 9.81 | m/s$^2$ |
| cylinder diameter | 0.025 | m |
| cylinder height | 0.025 | m |
| cylinder filling ratio | 0.3 | |
| particles' material density | 7800 | kg/m$^3$ |
| friction coefficient | 0.3 | |
| particles and wall Young's modulus | 200 | GPa |
| particles and wall Poisson's ratio | 0.3 | |
| collision restitution coefficient | 1.0 | |
| simulation time | 1.0 | s |

The contact model used by Abaqus Explicit, Chrono SMC, and Chrono GPU computes the normal force between the contacting bodies using Hertz's model; this model requires Young's modulus and Poisson's ratio as data. Therefore, to conform PMB to Abaqus and Chrono SMC models, the value of the normal contact linear spring value used in PMB has been chosen using a relation, dependent on Young's modulus and suggested in [33], for the equivalent maximum strain energy of the contacts

$$k_{n,mat} = 1.053 \left( \dot{\delta}_0 \, m'^{\frac{1}{2}} \, R' \, E'^2 \right)^{\frac{2}{5}},$$ (12)

where $\dot{\delta}_0$ is the estimated impact speed, $m'$ the effective mass, $R'$ the effective radius and $E'$ the effective Young's modulus.

Different DEM methods and contact models led to different time steps for each code. Table 3 reports the different codes time step, as a function of the particle diameters.

**Table 3.** Employed particle radius and average time steps. Values of contact normal stiffness used in PMB, for both the contact models, are also reported. Blank cells are left for those analyses not carried out because they are too time consuming.

| Particle Diameter, mm | 3.0 | 2.0 | 1.5 | 1.0 | 0.75 | 0.50 | 0.25 | 0.15 |
|---|---|---|---|---|---|---|---|---|
| Number of particles | 151 | 602 | 1569 | 5200 | 12,383 | 41,203 | 328,435 | 1,543,449 |
| Abaqus Expl. $\Delta T$, μs | 1.46 | 0.971 | 0.705 | 0.419 | 0.298 | 0.199 | | |
| Chrono SMC $\Delta T$, μs | 1.2 | 0.9 | 0.7 | | | | | |
| Chrono NSC $\Delta T$, μs | 10 | 10 | 10 | 10 | | | | |
| Chrono GPU $\Delta T$, μs | 1.2 | 0.9 | 0.7 | 0.4 | 0.3 | 0.2 | 0.1 | |
| PMB (both contact models) $\Delta T$, μs | 1.2 | 0.9 | 0.7 | 0.4 | 0.3 | 0.2 | 0.1 | 0.08 |
| PMB $k_n$ (between particles), N/m | $1.182 \times 10^6$ | $7.883 \times 10^5$ | $5.912 \times 10^5$ | $3.941 \times 10^5$ | $2.956 \times 10^5$ | $1.971 \times 10^5$ | $9.853 \times 10^4$ | $5.912 \times 10^4$ |

The results obtained using PMB, Chrono GPU, Abaqus Explicit, and Chrono SMC showed an excellent correspondence in terms of the position of the center of mass of the particles, while Chrono NSC showed different results, cfr. Figure 4. The comparison of the energy dissipated per period, not reported, led to the same observations. The difference in the results is due to the non-smooth method employed in Chrono NSC, which leads to a different dissipation; this is likely because the algorithm for solving cone complementarity problems for non-smooth dynamics maximizes the tangential dissipation. Such effects should be further investigated in future works.

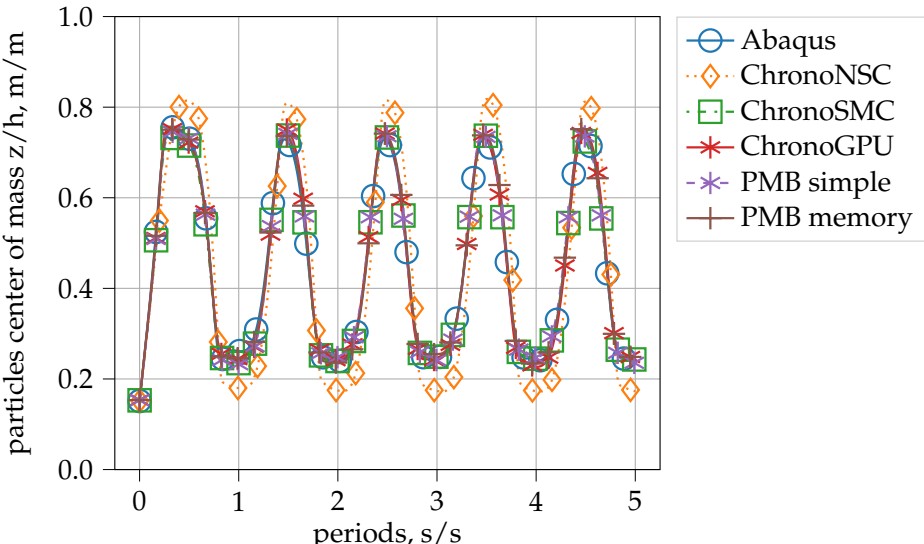

**Figure 4.** Center of mass trajectory of the particles in the DEM's validation case study of section for particles of diameter 3 mm.

The execution times of the analyses are compared in Figure 5, showing that the new software is faster than the other solvers. This comparison was obtained with simulations where only the total force of the impacts acting on the inner surface of the particle damper was computed and written to file at each time step.

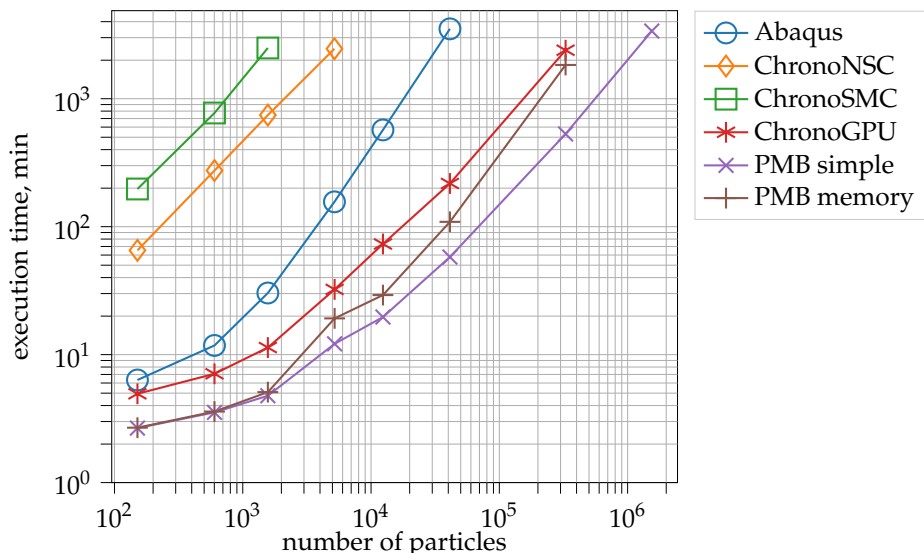

**Figure 5.** Execution times of the DEM's validation case study for increasing number of particles. Hardware: Intel(R) Core(TM) i7-3930K CPU @ 3.20GHz, Nvidia GeForce GTX 1650 4 GB GDDR6.

PMB with the simple friction model is also three times faster than Chrono GPU for most of the analyses; such speed was mainly obtained because PMB's simple friction model

does not require storing in memory contact parameters at each time step, while Chrono GPU can use more complex and physical friction models such as the Mindlin-Deresiewicz model. Additionally, the analyses performed with PMB with the elastic-frictional model are faster than those performed with Chrono GPU; however, increasing the number of particles simulated, the difference in execution time decreased. Chrono GPU would likely become faster for a higher number of particles; these results are likely due to different implementations of the broad-phase contact detection, different memory management, and different handling of the contacts' internal variables.

### 3.2. Fast Shear Test with the Simple Directional Friction Model

The purpose of this test is to evaluate whether the use of a simple directional friction model impacts the results of the simulation of a fast shear test, and hence if the particle bed retains the ability to respond to shear deformation, despite the use of a model lacking static friction.

This test was inspired by Fleischmann's work [51], which shows that elastic-frictional models do not show static friction in a shear test if the tangential displacement is computed only in the last step, without considering an accumulation of tangential displacement along all the time steps the bodies passed in contact.

The test consists of a numerical simulation of a shear test performed at high speeds. The simulations were performed with PMB using both the simple directional friction model and the elastic-frictional model. A high value of shear speed was selected because it was more consistent with particle damping conditions. It is also known that the use of the simple directional model for quasi-static deformations provides the wrong results because the simple model cannot model static friction between the particles, and also because an unphysical continuous packing of the particle bed would occur in quasi-static conditions.

The initial arrangement of the particles is random. For this reason, the tests were repeated 50 times and the results were averaged to reduce the deviations due to the initial random configuration. To efficiently arrange the particles in random positions, an iterative procedure using PMB's GPU-based loose uniform grid broad-phase was developed. The particles were assigned random positions fitting in the admissible domain without touching the walls and they were checked for inter-particle contacts; particles in contact were flagged to be reassigned to a new random position in the following iteration. The iterative procedure ended when all the particles had been assigned a position without touching other particles, or when a maximum number of iterations was reached.

The shear box used in the simulation has a cuboid shape. In PMB, the shear box was modeled as an "Assembled" enclosure, in which all the surfaces comprising the enclosure can be moved and oriented independently using runtime compiled laws. The ceil of the upper container was initially lifted at 10 times the total height of the shear box; the particles were deployed at random positions under the lifted ceil and left to fall under gravity inside the proper shear box. At time $t_1$, the lifted ceil started to move toward the reference position of the shear box's ceil, which was reached at time $t_2$. Between $t_3$ and $t_4$, the ceil and floor of the shear box moved symmetrically to compress the particles, which were then left to consolidate until $t_5$, which was the starting time of the shear motion of the upper container, which moved for 10 mm at a speed of 100 mm/s.

Whenever particles are modeled with a simple directional friction model, they continue to consolidate their position until reaching a very packed arrangement, which takes a long time. For this reason, the actual time history of the imposed motion of the shear box's container affects the results, which can be reproduced only by following the same packing procedure.

Table 4 reports the parameters used in the shear test simulations.

**Table 4.** Dimensions of the fast shear test simulations.

| Description | Value | | Description | Value | |
|---|---|---|---|---|---|
| Repetitions | 50 | | $t_4$ | 1.8 | s |
| Box shear tot height | 0.04 | m | $t_5$ | 2.5 | s |
| Box shear width | 0.12 | m | Particles Number | 5000 | |
| Box shear length | 0.12 | m | Particles Diameter | 0.005 | m |
| Initial ceil lifting | 0.7 | m | Particles Density | 2500 | kg/m$^3$ |
| Total compression | 0.005 | m | Contact Stiffness | 62,949.9 | N/m |
| Shear displacement | 0.01 | m | Equiv. Young's Mod | $8 \times 10^8$ | Pa |
| Shear velocity | 0.1 | m/s | Particles Friction Coef. | 0.5 | |
| $t_1$ | 1.0 | s | Shear Box Walls Friction Coef. | 0 | |
| $t_2$ | 1.2 | s | Gravity | 9.81 | m/s$^2$ |
| $t_3$ | 1.5 | s | | | |

The total force acting on each of the surfaces composing the shear box was computed and written to an output file throughout the simulation. The forces acting on the surfaces of the two containers of the shear box were separately summed. Figure 6 shows the ratio between the shear force and the normal force computed in PMB's analyses and averaged over the 50 different simulations. The shear force is here defined as the mean between the shear resultant modulus exchanged on the upper container and the shear resultant modulus exchanged with the lower container; the normal force is the mean between the absolute values of the vertical component of the force acting on the upper container and the one acting on the lower container, and is computed at time $t_5$.

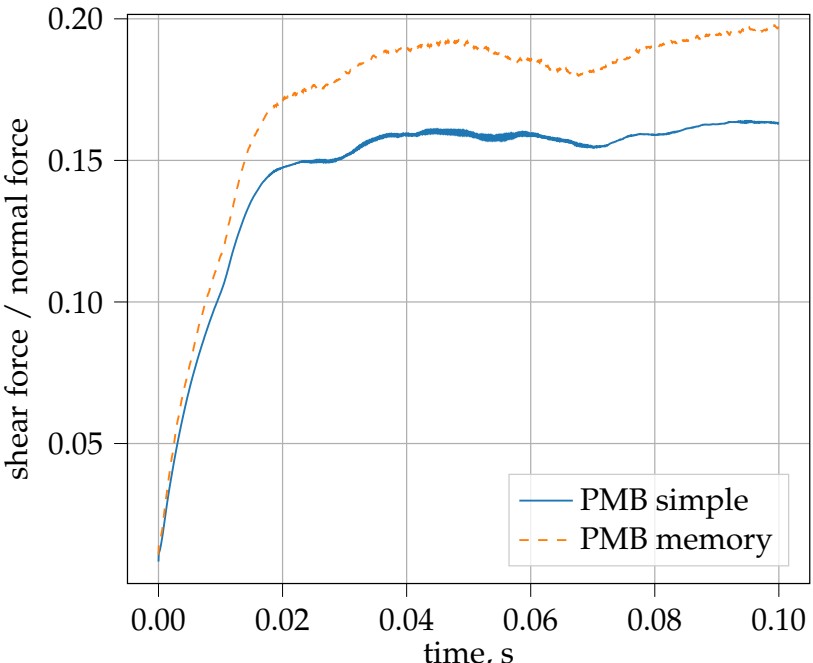

**Figure 6.** Ratio between the shear force and the normal force computed in PMB's analyses and averaged over the 50 repetitions of the analyses obtained in the fast shear test simulations.

The figure shows that the ratio between shear and normal forces obtained with the two friction models is similar, but not equal; therefore, this test case shows that the use of a simple directional friction model did not compromise the ability of the particle bed to respond to shear deformations, even if the response obtained with this model does not correspond with precision to that obtained with the elastic-frictional model.

### 3.3. An Application of the Contact Stiffness Sizing Technique

As explained in Section 2.2, the definition of the minimum value of normal contact stiffness ensuring accurate numerical results requires knowledge, with enough confidence, of the constants $\beta$ and $\eta$. Many simulations were performed, for different values of the contact stiffness, to estimate suitable values for those constants.

The model represents a mass–spring–damper system subjected to an imposed sinusoidal motion of the ground and with attached a particle damper, as in Saeki's experiment [43]; cfr. Figure 5b of the aforementioned paper for the original experimental results. To assess whether or not the reduction of contact stiffness affects the accuracy of the results, an error index was chosen to compare the results of one configuration for different values of contact stiffness; the selected index is the $l^1$-norm of the relative difference between the transfer function of the system obtained using a contact stiffness related to the materials' properties and that obtained with the model using a scaled contact stiffness, divided for the number of points in the transfer function. The set of contact stiffnesses to be simulated for each configuration ranges from the one computed from the material characteristics to values that are lower by several orders of magnitude.

Table 5 shows the parameters used for the different configurations, and the value of $\beta$ and $\eta$ found when looking at each individual configuration.

**Table 5.** Set of configurations used to compute the minimum value of normal contact stiffness. The tables shows the parameters defining different configurations, and the value of $\beta$ and $\eta$ found when analyzing each individual configuration. (*L*—cavity length; *H*—cavity height; *A*—input amplitude; $d_P$—particle diameter; *n*—particles number; $\rho$—particles density; $\mu_f$—friction coefficient; *e*—restitution coefficient).

| Brief Description | *L*, mm | *H*, mm | *A*, mm | $d_P$, mm | *n* | $\rho$, kg/m³ | $\mu_f$ | *e* | $\eta$ | $\beta$ |
|---|---|---|---|---|---|---|---|---|---|---|
| Saeki's original | 58 | 38 | 1 | 6 | 200 | 1190 | 0.52 | 0.896 | 0.11 | 2.7 |
| Bigger particles | 58 | 38 | 1 | 12 | 25 | 1190 | 0.52 | 0.896 | 0.3 | 3.0 |
| Smaller particles | 58 | 38 | 1 | 4 | 625 | 1190 | 0.52 | 0.896 | 0.1 | 2.8 |
| Capsule (*L*: height, *H*: diameter) | 58 | 38 | 1 | 6 | 200 | 1190 | 0.52 | 0.896 | 0.1 | 3.0 |
| Double amplitude | 58 | 38 | 2 | 6 | 200 | 1190 | 0.52 | 0.896 | 0.4 | 2.5 |
| Half $\mu_f$ | 58 | 38 | 1 | 6 | 200 | 1190 | 0.26 | 0.896 | 0.25 | 2.8 |
| A quarter $\mu_f$ | 38 | 38 | 1 | 6 | 200 | 1190 | 0.13 | 0.896 | 0.2 | 3.0 |
| Heavier particles | 58 | 38 | 1 | 6 | 200 | 4760 | 0.52 | 0.896 | 0.1 | 2.3 |
| Longer cavity | 78 | 38 | 1 | 6 | 200 | 1190 | 0.52 | 0.896 | 0.3 | 2.3 |
| Shorter cavity | 38 | 38 | 1 | 6 | 200 | 1190 | 0.52 | 0.896 | 0.8 | 2.6 |
| Gravity at 45 deg wrt motion | 58 | 38 | 1 | 6 | 200 | 1190 | 0.52 | 0.896 | 0.11 | 2.8 |
| Gravity parallel to motion | 58 | 38 | 1 | 6 | 200 | 1190 | 0.52 | 0.896 | 0.11 | 2.8 |
| | | | | | | | | | 0.1 | 3.0 |

The estimation of $\beta$ and $\eta$ for each individual configuration was achieved by graphically comparing the computed transfer functions, maximum overlaps, and contact stiffness. For instance, with reference to Saeki's original configuration described in Table 5, $\beta$ and $\eta$ were estimated using Figures 7 and 8a,b. Figure 7 compares the computed transfer functions for Saeki's experiment [43] (Figure 5b) simulated with different contact stiffnesses; clearly, for increasing values of contact stiffness, the numerical transfer functions tend to the reference one. Figure 8a shows the error index plotted against the maximum overlaps computed in the simulations; it is evident that as long as the computed overlap stays below a threshold equal to 11% of the particle radius, the error index is limited. Figure 8b shows, for analyses with different contact stiffness, the ratio between the computed maximum overlap over the reference overlap defined in Equation (8); that ratio never exceeds 2.7, as long as the maximum overlap is under the threshold found in Figure 8a .

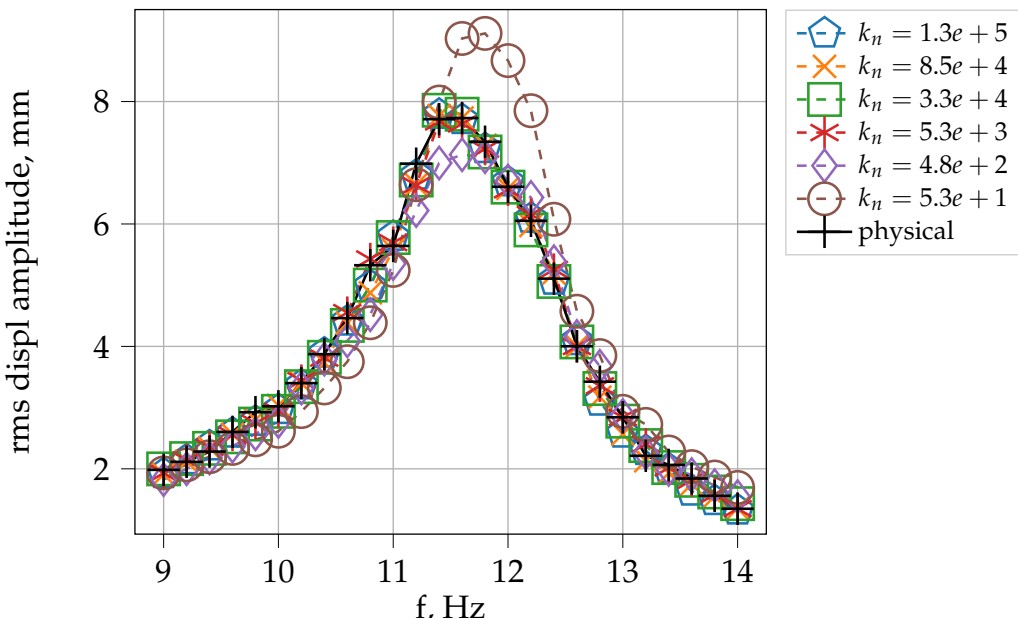

**Figure 7.** Transfer functions to reproduce Saeki's experimental results of Figure 5b [43] for different contact stiffnesses, compared to the reference transfer function, labeled in the legend as "material", obtained with a contact stiffness obtained from the material properties (particle to particle: $k_{n,P2P} = 9.06 \times 10^4$ N/m, particle to wall: $k_{n,P2W} = 1.37 \times 10^5$ N/m)

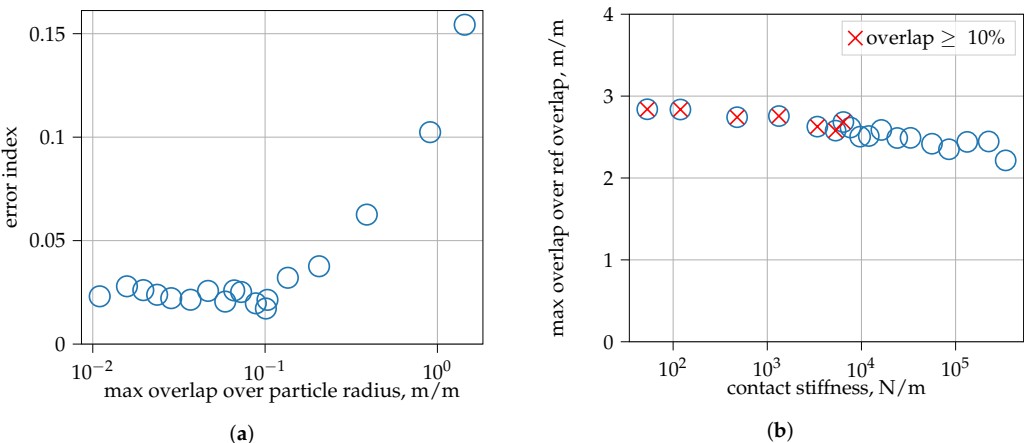

**Figure 8.** Figures for the estimatation of $\beta$ and $\eta$ relative to the simulation of Saeki's experiment of (Figure 5b [43]) for different contact stiffnesses. (**a**) Similarity index plotted against the maximum overlaps computed in the simulations, estimation of $\eta$. (**b**) Ratio between the computed maximum overlap over the reference overlap, estimation of $\beta$.

As shown in Table 5, $\beta$ never exceeded 3.0, and $\eta$ never reached below 0.1 for any different configuration; therefore, $\beta = 3$ and $\eta = 0.1$ are likely to be safe values in order to compute the normal contact stiffness with Equation (7), as long as they are employed for analyses with particle damper's working conditions that are not significantly different from those considered in this subsection.

### 3.4. Testing the Described Techniques for Wide Ranges of Amplitude and Frequency Conditions

The analyses in this section aim to test how the use of a simple friction model and the application of the contact stiffness sizing technique of Section 2.2 impact the numerical estimation of the damping characteristics of a particle damper over a wide range of motion frequencies and amplitudes.

The dimensions and material characteristics of the particle damper used in the following analyses are taken from Yin's work [52] (table 1); the particle damper is a cylindrical cavity subjected to a sinusoidal vertical motion and filled with 500 particles.

Yin performed the analyses using the Altair EDEM solver. Some of his analyses were reproduced with PMB, and particles' motion regimes and energy dissipation per period were compared with those reported in [52]. It was noted that

- The motion regimes are similar to those identified by Yin; good correspondence was obtained for bouncing bed and fluidization regimes, but the Leidenfrost effect and the buoyancy convection regimes were not always identified in the same conditions reported by Yin; a possible reason for this discrepancy could be that the definitions reported by Yin for those regimes are somewhat subjective;
- Excellent correspondence in energy dissipation per period was found for the bouncing bed regime only. Unfortunately, the other motion regimes are characterized by dissipations that change significantly from one cycle to another without converging to a single value, and Yin does not refer to this behavior.

The main results of this section were obtained over a set of analyses performed on the aforementioned particle damper consisting of imposed vertical sinusoidal motion of different frequencies and intensities (maximum accelerations in g). An array of 13 frequencies per 21 intensities was simulated; the considered frequencies are 5, 6, 7.5, 10, 12.5, 15, 20, 25, 35, 50, 70, 90, 100 Hz. The 21 intensities are logarithmically spaced from a minimum value to a maximum value; the minimum value was set to 0.8 g regardless of the motion's frequency, while the maximum value is set to 10 g for a frequency of 5 Hz, and increases linearly to 100 g for a frequency of 100 Hz.

The analyses were repeated both with the simple directional friction model and the elastic-frictional model, and with different normal contact stiffnesses, computed either:

- From the elastic material properties;
- Using the method of Section 2.2, with a contact duration at least 100 times shorter than the oscillation period, $\beta = 3$, $\eta = 2\%$, $\eta = 5\%$ and $\eta = 10\%$;

Each analysis was run for several cycles. The energy dissipation for each of the last ten cycles was computed, and used to compute the particle damper equivalent damping ratio, for a reference mass ratio $\mu = 1\%$, as shown in Section 2.3. Therefore, the quantity $\xi_{PD}(1\%)$ is used to compare the damping performance of the particle damper in different conditions.

3.4.1. Damping Performance and Standard Deviation Using Material Properties and Elastic-Frictional Model

Since the response of a particle damper is not guaranteed to converge to a periodic response, the energy dissipation per cycle, and the corresponding equivalent damping ratio, should be discussed in statistical terms, e.g., mean value and standard deviation. Figure 9a,b show the contour plots of the mean and the relative standard deviation of the equivalent damping ratio $\xi_{PD}(1\%)$ of the last ten periods of each simulation in the frequency-intensity array, obtained using the material properties for the contact stiffness and the elastic-frictional model. Two main damping zones can be identified in the contour of the mean equivalent damping ratio (Figure 9a):

- The bouncing bed zone, in correspondence with a motion amplitude greater than about half the clearance inside the particle damper (greater than 0.015 m);
- The initial fluidization zone, in correspondence of intensities between 1.5 g and 4 g; this zone does not depend on frequency.

The same two zones can also be found in the literature, even for particle dampers oscillating in the horizontal direction. Furthermore, Figure 9b shows that the main damping zones are also characterized by low values of the relative standard deviation of the dissipation. It must be noted, however, that high values of relative standard deviation were found at low intensities, below or around 1 g; the particle bed in such conditions is mainly static and the total dissipation is low, and the low damping values may depend on small

rearrangements of single particles during the oscillations. High values of relative standard deviation were obtained outside the two damping zones, where the particles are unlikely to show a periodic response and a chaotic response often emerges.

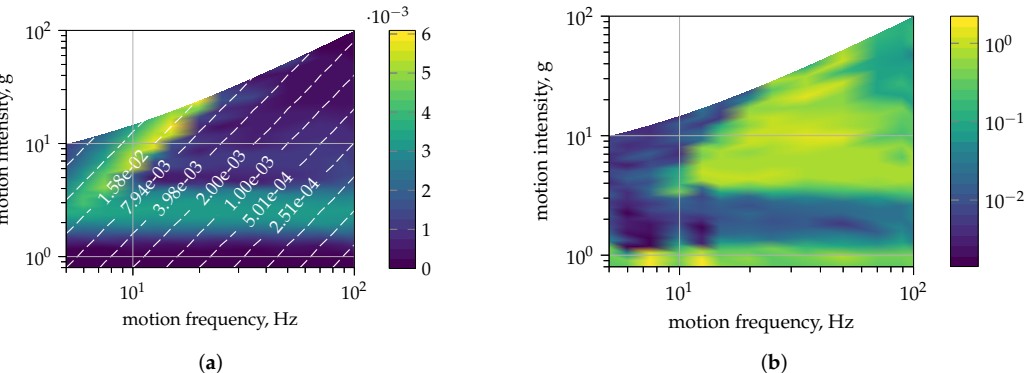

**Figure 9.** Mean and relative standard deviation of the equivalent damping ratios $\xi_{PD}(1\%)$ obtained from the last ten periods of each simulation. Using material properties and elastic-frictional model. (**a**) Mean equivalent damping ratio, white contour is the motion amplitude in m. (**b**) Equivalent damping ratio relative standard deviation.

3.4.2. Influence of the Friction Model and Stiffness Softening on the Mean Equivalent Damping Ratio

In this section, the equivalent damping ratios obtained with different contact models and properties are compared in order to evaluate the effect of a simpler friction model and softer contact stiffnesses on the estimation of the mean equivalent damping ratio. The mean equivalent damping ratios obtained with every contact property and model configuration analyzed qualitatively shows the same main damping zones shown in Figure 9a. In order to quantitatively analyze the differences in damping performance, the relative difference between different contact properties and model configurations are reported.

Figure 10a shows the relative difference of the mean dissipation per cycle (obtained from the last 10 cycles of each simulation) between simulations performed with the simple directional friction and the elastic-frictional model, using normal contact stiffness derived from the material's properties. Low relative differences were found in the two main damping zones; however, significant relative differences were found for intensities lower or equal to 1 g and outside the main damping zones. Significant relative differences outside the main damping zones are not of concern: outside the damping zones, the standard deviation of the dissipation is substantially high and the damping ratio is low, therefore the absolute difference in damping in such zones is limited.

Figure 10b shows the relative difference in mean dissipation per cycle between simulations performed using normal contact stiffness derived from the material's properties, and the normal contact stiffness obtained with $\eta = 2\%$. Even if this figure is quite similar to Figure 10a, the relative differences in dissipation in the damping zone at medium-high frequencies (>20 Hz) are more significant, even if limited. The contact stiffness in those conditions was determined by the constraint on the maximum duration of the impacts; possibly such constraint is not strict enough to reproduce the simulations with more accuracy in those conditions.

Figure 10c,d refer to the relative difference for $\eta = 5\%$ and $\eta = 10\%$; the difference in dissipation also increased in the bouncing bed damping zone. Nevertheless, the difference remained limited (<10%) for low frequencies and intensities, similar to those analyzed in Section 3.3.

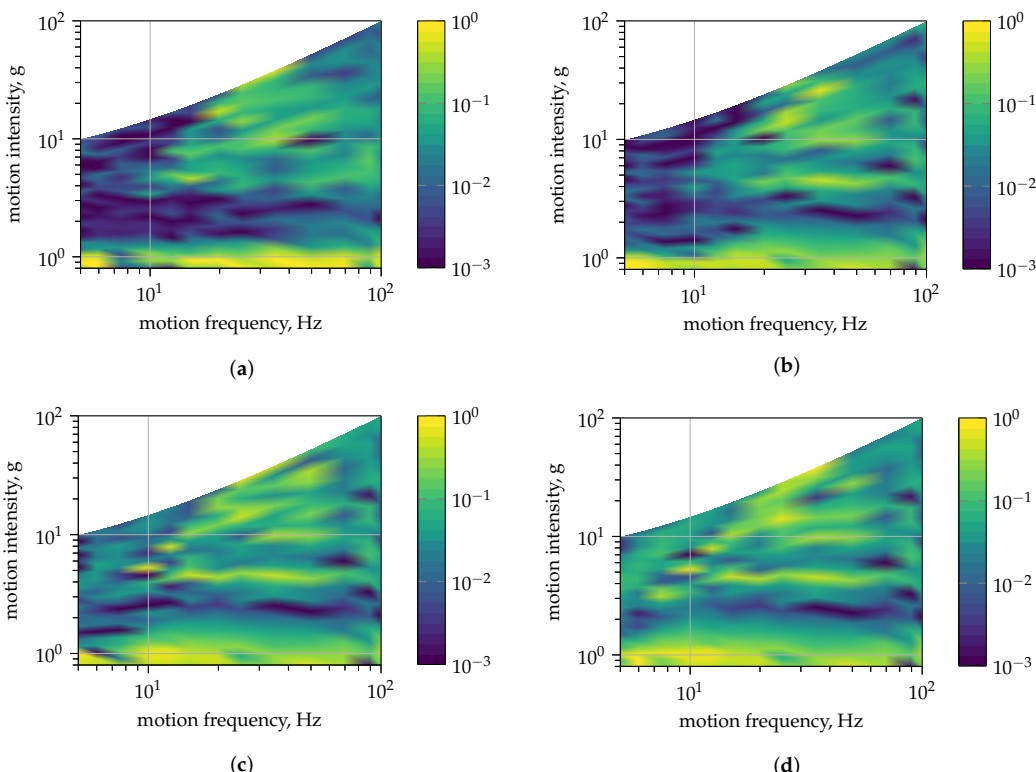

**Figure 10.** Point by point relative difference, relative to the greater of the two, between mean dissipations of the last 10 periods obtained with different model and normal stiffness configurations. (**a**) Dissipation relative difference between simulations with the simple directional friction model and the elastic-frictional model. Contact stiffness derived from material properties. (**b**) Dissipation relative difference between simulations with contact stiffness derived from material properties and stiffness obtained for $\eta = 2\%$. Elastic-frictional model. (**c**) Dissipation relative difference between simulations with contact stiffness derived from material properties and stiffness obtained for $\eta = 5\%$. Elastic-frictional model. (**d**) Dissipation relative difference between simulations with contact stiffness derived from material properties and stiffness obtained for $\eta = 10\%$. Elastic-frictional model.

Finally, the results of this section show that using a simple directional friction model has little influence on the estimation of the damping ratio over a large range of frequencies and intensities for this particle damper. Furthermore, the selection of a lower contact stiffness until $\eta = 10\%$ affected the accuracy of the estimation of damping, but apparently did not affect the position of the main damping zones.

## 3.5. Experimental Results and Numerical Correlation

In this section, the results of the experiment described in Section 2.4 and the numerical correlation with PMB coupled to MBDyn are presented. The results accounting both for the damping performance and the particles' motion regimes are compared.

### 3.5.1. Experimental Results

For each level of imposed acceleration at the free end, the transfer functions between the input force and the accelerometers mounted on the beam were measured, as described in Section 2.4.

The half-power bandwidth method was used to estimate the critical damping ratios of those transfer functions around the first bending mode. Figure 11 is a tridimensional plot comparing the aforementioned measured transfer functions at different amplitude levels and the corresponding damping ratios. The shape of the damping curve is characterized by low values of damping for low levels of imposed acceleration and a peak in damping at 1.25 g.

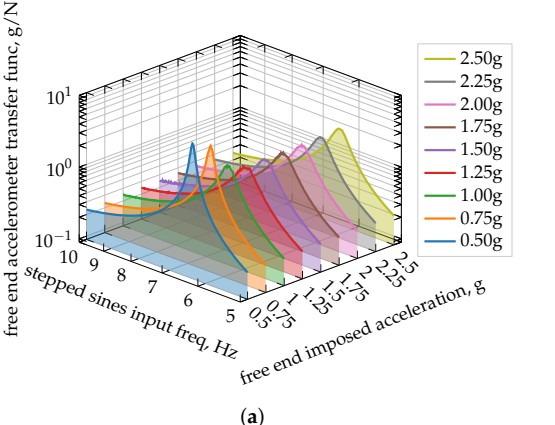
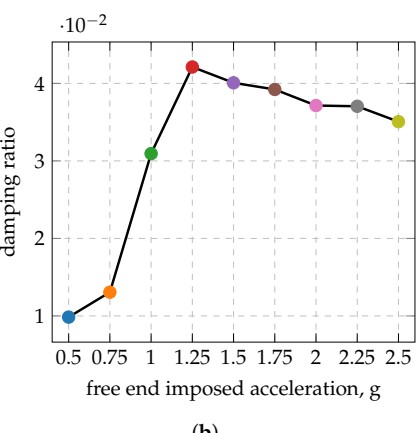

(**a**)  (**b**)

**Figure 11.** Experimental transfer functions and damping ratios per acceleration level, referred to the accelerometer near the free end. (**a**) Experimental transfer functions. (**b**) Damping ratios per acceleration level.

Slow-motion videos were visually analyzed to qualitatively identify the particles' motion regime at different positions on the damping curve; the cameras captured the experiment both from above and from the side. The following observations were made:

- At 0.5 g and 0.75 g, only a few particles in the upper layer are moving. Excluding those particles, the particle bed appears to be globally at rest.
- At 1.0 g and 1.25 g, all the particles in the upper layers are in motion in the cavity. Using Meyer's terminology [6], the particle bed is locally fluidized in the upper layers.
- Starting from 1.5 g, all the particles are involved in the global motion of the particles' bed (global fluidization). The particles collapse on the cylinder's planar sides leaving a free surface of the particle bed that is significantly oblique with respect to the planar sides themselves.
- Increasing the imposed acceleration, the free surface of the particles tends to become more parallel to the planar sides of the cylindrical cavity; hence, the motion regime transitions to a bouncing bed regime.

3.5.2. Numerical Correlation

Two correlation phases are required; in the first phase, the response of the underlying structure without particle damper is matched, while the second phase allows the particle damper's contact parameters to be tuned starting from values derived from the colliding materials' elastic properties.

Particle dampers are sensitive to the amplitude of the oscillations; thus, the elastic, mass, and damping properties of the bare structure without particles need to be updated so that the numerical frequency response and the modal masses and stiffnesses of the system match the experimental ones. Failing to perform this step could compromise the estimation of the damping performance of the system, as the mass ratio of the modeled particle damped system, which is one of the most influential parameters in particle dampers, could not correspond to the experimental one.

The contact parameters' initial values had to be selected from literature data. The sliding friction coefficient and restitution coefficient values are not uniquely defined: the friction coefficient relative to steel on steel sliding varies between 0.4 and 0.6, while the restitution coefficient is also dependent on the impact velocity. PMB only considers constant restitution coefficients; thus, a single value needs to be selected. Data from literature surveys [53] show that the restitution coefficient can be assumed to be equal to 0.9 for the ranges of velocities at hand (0.1–1.0 m/s). The rolling friction coefficient largely depends on the geometric imperfections of the particles.

The contact stiffness between the particles, and between the particles and the enclosure's walls, was defined using the contact stiffness sizing technique in Section 2.2. The

values $\beta = 3$ and $\eta = 0.1$, estimated in Section 3.3, were obtained in conditions quite similar to those in this experiment; the same values are thus adopted here. To confirm the validity of such approximation, the simulation of the experiment for an imposed acceleration equal to 1.0 g was repeated using different values of stiffness obtained with $\eta$ ranging from 0.033 to 0.8; the obtained response of the numerical system converges for $\eta$ lower or equal than 0.1.

Since the friction coefficient and the rolling friction coefficient are not uniquely defined in the literature, the calibration of these parameters must be performed based on the experimental results; by repeating the analysis of a given experimental condition, e.g., at an acceleration of 1.0 g, it was found that these friction parameters do not substantially influence the results on the analyzed conditions, provided their value is reasonable and consistent with the material at hand. However, a sliding friction coefficient of 0.6 and a rolling friction coefficient of 0.01 are needed to achieve good correspondence between the experimental and the numerical response when the particle bed is globally fluidized.

For reasons of computational efficiency, the simple directional friction model was employed in the simulation of this experiment.

### 3.5.3. Comparison of the Experimental and Numerical Results Obtained

Figure 12a compares the experimental and the numerical damping ratios estimated by applying the half-power bandwidth method on the transfer functions for each considered imposed acceleration amplitude of oscillation at the particle damper.

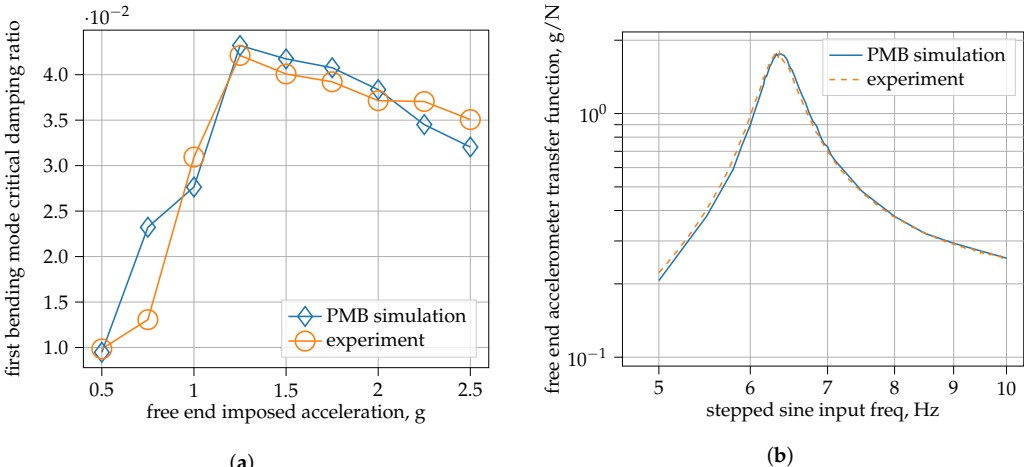

**Figure 12.** Comparison between the results of the experiment of Section 2.4, and the results of the numerical analyses using PMB coupled to MBDyn. (**a**) Comparison between the damping ratios computed with the half-bandwidth method. (**b**) Comparison between the transfer functions for an imposed acceleration equal to 2 g.

For high values of the imposed accelerations, the computed experimental damping ratios are similar; the damping curves correspond well for accelerations greater than 1.0 g. This correspondence was also obtained in the transfer functions, which are almost overlapping in most of the simulated cases; see the transfer function obtained for an imposed acceleration of 2 g and reported in Figure 12b.

Lower values of the imposed acceleration, equal to 0.75 g, correspond to a regime close to the onset of partial fluidization of the particles' bed. In this case, significantly different damping ratios were obtained. This is because the particles in the simulation experienced higher mobility than in the experiments; the experimental videos show that only a few particles are moving, while the simulations' replay videos show that the upper layers of the particles' bed are subject to partial fluidization; hence, the simulation predicts higher dissipation. Figure 13 shows a frame of the simulation at 0.75 g for the case of the horizontal oscillation and particle damper's axis aligned to the oscillation's direction. As shown by the instant velocity of the particles, the particles closer to the cylinder's axis of the damper have

higher velocity than those closer to the enclosure lateral surface, creating more energetic impacts and increasing the dissipation. Furthermore, during the iterative amplitude control procedure, the particles in the simulation arranged themselves in a hexagonal lattice near the lateral surface, leaving fewer particles near the center of the cavity; such arrangement was not seen in the experiments.

An initial explanation of this damping difference could be the simplified friction model, which does not account for static friction. The simulation with an imposed acceleration of 0.75 g was thus repeated both with PMB and the elastic-frictional model and with Chrono GPU, both coupled to MBDyn with a loose coupling scheme. The experimental damping ratio at 0.75 g, and the damping ratios obtained with PMB with the simple friction model, PMB with the elastic-frictional model, and Chrono GPU are 1.31%, 2.32%, 2.40%, and 2.32%, respectively. Therefore, the friction model does not seem to explain this difference with respect to the experimental results; the authors speculate that a possible reason for this difference could be the incidental arrangement of the particles occurring during the iterative convergence procedure in the simulations.

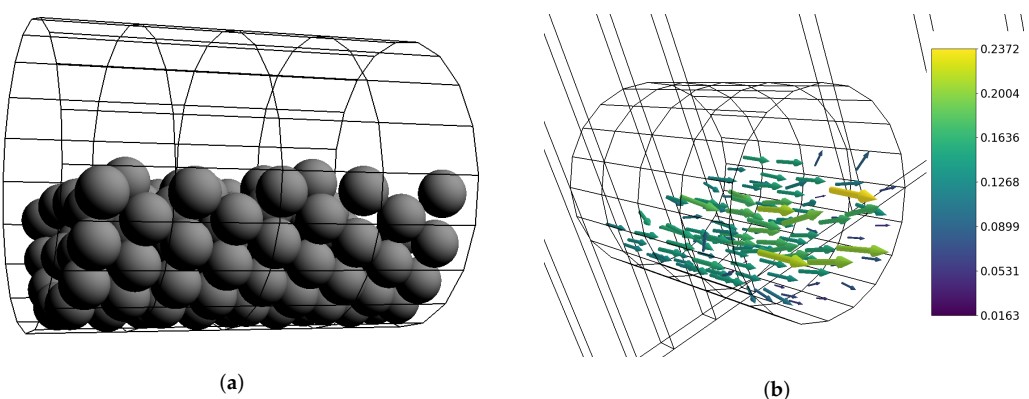

(**a**)                                                                                       (**b**)

**Figure 13.** Frames of the simulation at imposed acceleration amplitude 0.75 g at frequency 6.3 Hz of the simulation of the experiment horizontally oscillating, with the damper's axis parallel to the oscillation's direction. Values in the colorbar are in m/s. (**a**) Particles positions. (**b**) Particles velocities, Viridis colormap.

## 4. Conclusions

In this paper, two techniques to make the DEM simulation of particle dampers more efficient were proposed, and the impact of such techniques on the DEM estimation of the dissipation characteristics of particle dampers with spherical particles was evaluated.

A novel smooth DEM code tailored to particle dampers, called PMB, was presented and tested in comparison with other available solvers. PMB proved itself significantly more efficient than the compared solvers (Section 3.1); the other results in this paper were obtained within a reasonable time because of its efficiency and speed in simulating particle dampers. PMB was coupled to MBDyn using a loose coupling scheme; therefore, the effect of the application of a particle damper with spherical particles and rigid shape can be evaluated on any multi-body system that requires damping of vibrations.

The adoption of a simple directional frictional model, neglecting static friction, provided a sensible reduction in simulation execution times with respect to an elastic-frictional model that requires the memorization of the contacts and internal variables.

Contrary to that which was shown by Fleischmann for single-step elastic-frictional models in quasi-static conditions, the use of a more efficient directional friction model at high speeds did not completely tamper with the particle bed's ability to respond to shear; the shear to normal force ratio was, however, lower when using the directional friction model: this result should be further investigated in future works.

The simulations performed on particle dampers using the directional friction model show a good correspondence in the estimation of the damping characteristics and the

particles' motion regimes predicted with experiments or simulations with more complex friction models:

- The simulations in the simple case study of Section 3.1 give the same results, in terms of the position of the particles' center of mass and energy dissipation, with PMB's simple friction model, PMB's elastic-frictional model, and other DEM solvers;
- The simulations of Yin's particle damper [52] in Section 3.4, were performed over a wide range of frequencies and intensities of an imposed sinusoidal motion of the particle damper and showed that the adoption of the simple directional model does not necessarily significantly alter the overall particle damper's dissipation, provided that the particle dampers working condition is within one of the two main damping zones. In the author's opinion, the differences found for the other oscillation conditions are not a significant limitation since, as shown in Section 3.4.2, not only the actual value of damping obtained for those conditions is relatively small, but the high variance of dissipation per period in the same simulation hints to a chaotic response and an undesirable particle damper's working condition.

The use of a simplified friction model appears to not influence the results of particle dampers simulations for all the selected case studies.

A technique to reduce the normal contact stiffness in the simulation of particle dampers and estimate the minimum value of contact stiffness that does not affect the accuracy of the numerical results was proposed; as shown in Section 2.2, this technique can reduce the execution time of particle dampers simulations by several times. This technique is based on the assumption that the stiffness of the contacts can be reduced as long as the overlap between the particles does not exceed a certain threshold, referred to as $\eta$. An example of the application of such a technique is shown in Section 3.3; this leads to the definition of the constants $\beta$ and $\eta$ to compute the minimum value of contact stiffness (Equation (7)) for different configurations of particle dampers coupled to a deformable mechanical system.

The influence of the reduction in contact stiffness was also studied for Yin's particle damper, considering a wide range of frequencies and intensities. The results show that overlap as high as $\eta = 2\%$ of the particles' radius does not significantly influence the dissipation performance in the two main damping zones; as the allowable overlap threshold $\eta$ increases, the estimation of the damping performance is increasingly affected. Therefore, one can conclude that the contact stiffness reduction can provide significant time savings, but should be applied with care and only after calibration to analyze its impact on the results.

Finally, a new experiment consisting of a cantilever beam with a particle damper near the free end oscillating in the horizontal direction at controlled acceleration RMS was considered. The simulations required

- The implementation of specific numerical amplitude control techniques, created to obtain the response of highly non-linear dynamic systems, such as particle dampers attached to multibody systems or flexible structures; such techniques speed up the convergence of the response and, more significantly allow for the reduction of the number of periods that needs to be simulated before moving to the next frequency;
- A two-phase correlation procedure to update the model of the deformable structure and select the most suitable contact parameters for the analysis, spanning the definition of the normal contact stiffness with the definition of the right value of $\eta$ and $\beta$ and the correlation with the friction and restitution parameters.

Despite some inconsistencies found in the simulation results with the beam oscillating in the horizontal direction for target acceleration values close to those leading to partial fluidization, good correspondence with the experimental results was found in both the transfer functions and the comparison with videos of the experiment.

Additional analyses were performed, both using PMB and Chrono GPU, in order to check whether the differences were because of the simple directional friction model, and seem to rule out the simplified friction model as the root cause of the differences.

Depending on conditions such as amplitude and frequency of the particle damper's motion, the duration of the analyses may be scaled down by a factor higher than 10, see Figure 2; for example, the case study with more than a million of particle simulated, reported in Section 3.1, which required more than 2 days on consumer-grade hardware (Nvidia GeForce GTX 1650), could be simulated in only a few hours on the same hardware. Furthermore, for particle dampers involving only thousands of particles, many more analyses can be performed in the same time frame; depending on the specific case, the studied techniques could make extensive parameters and optimization studies of particle dampers more feasible.

**Author Contributions:** Conceptualization, M.M. and G.L.G.; methodology, F.B., M.M. and M.T.; software, F.B. and M.M.; validation, F.B.; formal analysis, F.B. and M.T.; investigation, F.B., M.M. and M.T.; resources, G.L.G., M.T. and P.C.; data curation, F.B. and M.T.; writing—original draft preparation, F.B., M.T. and M.M.; writing—review and editing, F.B. and M.M.; visualization, F.B.; supervision, M.M., G.L.G. and P.C.; project administration, G.L.G.; funding acquisition, G.L.G. All authors have read and agreed to the published version of the manuscript.

**Funding:** The project leading to this application has received funding from the Clean Sky 2 Joint Undertaking (JU) under grant agreement No 687023. The JU receives support from the European Union's Horizon 2020 research and innovation programme and the Clean Sky 2 JU members other than the Union. .

**Institutional Review Board Statement:** Not applicable.

**Informed Consent Statement:** Not applicable.

**Data Availability Statement:** Numerical simulation results and figures data are available from the Zenodo repository https://doi.org/10.5281/zenodo.6678352 (accessed on 1 June 2022). Raw experimental data are available upon requet to the authors.

**Acknowledgments:** The authors thank Dan Negrut and Ruochun Zhang from University of Wisconsin—Madison for their openness in sharing valuable insights and for their suggestions.

**Conflicts of Interest:** The authors declare no conflict of interest.

## Abbreviations

The following abbreviations are used in this manuscript:

| | |
|---|---|
| DEM | Discrete Element Method; |
| GPU | Graphics Processing Unit; |
| LSD | Linear Spring-Dashpot; |
| DOF | Degree Of Freedom. |

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
