# Peer review of "Efficient Discrete Element Modeling of Particle Dampers"

_processes, doi:10.3390/pr10071247_

Round 1

Reviewer 1 Report

This study investigated particle dampers with the experiment consists of a rectangular-section aluminum alloy cantilever beam and a cylindrical particle damper positioned near the free end, and the simulation between PMB and MBDyn. And the reduction in contact stiffness is also studied in relation to the maximum overlap between particles and to the contacts’ duration. This reviewer recommends a major revision to address the points below:

1.      In section 2.1.1, you said the PMB solves both the rotational and linear equations of motion using the semi-explicit Euler time integration scheme. In my opinion the explicit integration scheme is mostly used for DEM, so I think some explanation and references are needed here.

2.      In Section 2.1.1, more explanations are needed for the understand of the sentences “the enclosure can be oriented in any direction”.

3.      The expression of the Eq. 3 is confusing, especially for the conditions. The first equation means the linear spring dashpot method is used when the force is lower than the friction limit. But why the condition of the second equation is the velocity? Do you mean the condition of the second equation is that the particle velocity should be greater than a value? But the expression now seems like the velocity is calculated by the second equation. Please explain more.

4.      There is no explanation of the symbol VT,n used in Eq. 4.

5.      The Eq. 6 needs more explanations, such as the determination of the maximum overlap, or the meaning of the “didn’t cause any significant effect”. It will make a clearer understanding of the equation.

6.      I think it is possible to show the position of the damper in the Fig. 3a, which can make it more clearly to understand.

7.      The title of Section 2.4.2 is “Modeling of the experiment in PMB coupled to MBDyn”, but I only see the setting about MBDyn, but not about PMB.

8.      In section 3, you talked about different DEM methods and contact models, so I think the parameters you summarized in Table 2 are incomplete, such as the restitution coefficient and the Poisson ratio used in Hertz model.

9.      you said Fig. 4 showed an excellent correspondence in terms of the position of the center of mass of the particles and energy dissipated per period, but I think the energy dissipated per period can’t be get clearly from Fig. 4. Also, it is not obvious enough to stand out the different results obtained by Chrono NSC, which need more descriptions.

10.   In Section 3.2, you said “Figure 6 shows the ratio between the shear force and the normal force computed in PMB’s analyses and averaged over the 50 different simulations”, but in Fig. 6 it is shear stress but not the shear force, I think stress is different from force, please explain. It is also same for “normal force” and “normal stress”.

11.   You said “the force is the mean absolute value at time t5”. I think it is impossible to get an average value from a moment, perhaps what you want to say is the mean value during a time period.

12.   A schematic is needed for Fig. 13b, which indicates the speed value corresponding to the different colors.

Reviewer 2 Report

The article describes a novel approach to increase the simulation speed for the numerical simulation of particle dampers. The overall concept of the contribution is clear. However, there are also some questions should be addressed. I recommend this manuscript can be accepted after a mandatory revision. In order to improve the quality of the contribution, some suggestions for changes are listed below:

The abstract should be formulated more precisely. The following structure is suggested: Please modify it to show the background, the challenging, the focus of this study, the novelty and the meaningful results.

Some general applications of particle dampers have already been mentioned in the introduction. At this point it should also be mentioned that recently particle dampers can be realised by means of additive manufacturing. Some relevant sources are:

Scott-Emuakpor et al. https://doi.org/10.1016/j.addma.2020.101739

Ehlers at al. https://doi.org/10.1016/j.addma.2020.101752

Schmitz et al. https://doi.org/10.1016/j.precisioneng.2020.07.002#

Guo et al. https://doi.org/10.1016/j.powtec.2021.11.029

In particular, additive manufacturing poses the challenge that the particles are in the range of a few µm and the number of particles in the cavities is very high. Thus, the approach presented here can also be promising in the field of additive manufacturing, as the number of particles is very high.

The sentence from line 34 to 36 should be supported by a source. Numerical simulations can also be calibrated by adjusting material parameters etc.

In the introduction, statements should already be made about the number of particles and simulation time in order to better demonstrate the need for optimisation strategies. Perhaps there are pictures similar to Figure 5, which can be used here for motivation in the introduction. It is also not clear from the introduction how many particles are to be simulated in this paper. It is also not clear what requirements are placed on the modelling. In addition to the number of particles, the particle size and the frequency range investigated are relevant. These points should already be specified in the introduction.

In chapter 2 it is not clear why the MBDyn and PMB software are selected

In paragraph 184 to 190 it is said that the computing time must be further reduced in order to be able to carry out numerous parameter studies for the design. Depending on the number of particles, the simulation takes hours or days. At this point, a goal or research question should be formulated. This point should also be included at the end of the paper, whether the goal or research question could be realised. Up to what number of particles would the approach presented here be suitable as a design tool?

Layout Figure 2: First reference in the text then image. The exponents in the image could be omitted. The range of values is from 1 to 11.

It is not quite clear how Figure 2 is read. Why are the lines of 1.12e+01, 7.94e+00 constructed like a triangle?

Section 2.4.1: It should be mentioned why the particle damper is excited to vibrate in vertical direction and not in horizontal direction. It should be justified why steel balls and a diameter of 3 mm were chosen. It should also be explained why low frequencies are analysed and how the beam dimensions were determined.

Table 3: Table headings should be placed above the table. Furthermore, the designation of PMB simple and memory of Table 3 and Figure 4 should be identical.

Figure 7: What does "material" mean?

Line 494 and 495 Insert source.

Line 496: It should be mentioned how the videos were evaluated and where the camera was placed in the experimental setup. Furthermore, it is not clear in line 502 how the state of "global fluidization" could be detected with a camera when viewing from only one direction.

Honorary Authorship / Author Contributions: It should be clarified whether it is sufficient for "Potito Cordisco" to have only contributed to resources and supervision.

Round 2

Reviewer 1 Report

The authors have satisfactorily addressed the issues I raised in my review. I recommend the manuscript to be published on Processes.